# Antifungal Potency of Amaranth Leaf Extract: An In Vitro Study

**DOI:** 10.3390/plants12081723

**Published:** 2023-04-20

**Authors:** Agnieszka Jamiołkowska, Barbara Skwaryło-Bednarz, Radosław Kowalski, Ismet Yildirim, Elżbieta Patkowska

**Affiliations:** 1Department of Plant Protection, University of Life Sciences in Lublin, Leszczyńskiego 7, 20-069 Lublin, Poland; elzbieta.patkowska@up.lublin.pl; 2Department of Analysis and Food Quality Assessment, University of Life Sciences in Lublin, 20-950 Lublin, Poland; radoslaw.kowalski@up.lublin.pl; 3Department of Plant Protection, Faculty of Agriculture and Natural Sciences, Düzce University, 81000 Düzce, Turkey; ismetyildirim@duzce.edu.tr

**Keywords:** Amaranthus cruentus, Amaranthus hypochondriacus × hybridus, Amaranthus retroflexus, Amaranthus hybridus, plant extract, fungal diseases

## Abstract

Plant diseases are a serious problem for agricultural crops, the food industry and human health. Significant efforts have been made in recent years to find natural products that could reduce the growth of plant pathogens and improve food quality. At present, there is an increased interest in plants as a source of biological active compounds that can protect crops from diseases. Important sources of these phytochemicals are lesser-known pseudocereals such as amaranth. The objective of this study was to determine the antifungal activity of leaf extracts of four amaranth species (*A. cruentus*, *A. hypochondriacus* × *hybridus*, *A. retroflexus* and *A. hybridus*). The antifungal potency of amaranth extracts was analyzed against selected strains of fungi. The results suggested that the antimicrobial properties of the tested extracts varied depending on the amaranth species and the fungal strain. The studied extracts inhibited the growth of *Fusarium equiseti*, *Rhizoctonia solani*, *Trichoderma harzianum* and *Alternaria alternata*. A lower inhibitory effect of the extracts was recorded against *F. solani*, while no inhibitory effect was observed against *F. oxysporum* and *Colletotrichum coccodes*.

## 1. Introduction

Food-borne illnesses are a major concern for consumers, the food industry and food safety authorities. Pathogenic microorganisms present on plants cause diseases, as well as deterioration in the quality of stored food products. Among these microorganisms are many polyphagous phytopathogenic fungi that contribute to food spoilage. These include, among others, fungi of the genus *Fusarium*, *Botrytis* or *Alternaria*. In addition to the losses they cause in crops, they are also a source of mycotoxins that accumulate in food. The genus *Fusarium* produces specific toxins whose profile and harmfulness depend on the species and even strain of the fungus. Fusarium species are dangerous pathogens with a high toxicity potential. Secondary metabolites of these fungi, such as deoxynivalenol, zearalenone and fumonisin B1, are among the five most important mycotoxins in Europe and the world [1,2,3]. Similarly, the genus *Alternaria* can produce under certain conditions secondary metabolites with toxic properties, known as *Alternaria* toxins [4]. Currently, scientists are emphasizing the importance of several toxins, including alternariol (AOH), alternariol monomethyl ether (AME), altertoxin (ATX), altenuene (ALT), tenuazonic acid (TeA), tentoxin (TEN) and AAL toxins that can be found as contaminants in agricultural crops [5]. Mycotoxins can spread from rotten plant tissue to surrounding areas; thus, they are also present in processed foods [6]. Therefore, many pathogenic microorganisms remain an important problem for agriculture and the food industry. 

Consumers are concerned about synthetic preservatives used in food. As a result, there is a growing demand for natural products that can serve as an alternative to food preservatives. The current concept of sustainable agriculture assumes a reduction in the use of synthetic pesticides and a wider use of natural products for plant protection and food preservation [7]. Therefore, currently, natural preparations, which are as effective as chemical preparations, are increasingly used in integrated and organic production to protect plants against diseases [8,9]. Many of them are based on plant extracts enriched with minerals and beneficial microorganisms [10]. Numerous studies have been conducted on plants containing biologically active compounds that inhibit the growth of pathogenic microorganisms [11,12,13]. Many plant species, mainly herbs, are already promising sources of bioactive substances such as phenols, anthocyanins, flavonoids or carotenoids, which are used to extend the shelf life of meals and processed foods [14,15,16,17]. The action of these natural compounds is not specific, and their effects on pathogens vary. The natural bioactive compounds used in plant protection destroy pathogens (fungicides) or limit their development (fungistatics), as well as induce plant defense reactions by acting as elicitors [9]. Many studies have carried out systematic screenings of plants and the compounds contained in them with antimicrobial and antiviral properties. Europe is an area extremely rich in medicinal plants that contain a variety of antimicrobial components. In recent decades, the antimicrobial and antifungal properties of various extracts and their components, including essential oils, have been tested and attention has been drawn to the application of these natural raw materials as alternative plant protection products, because they are biodegradable and non-toxic to the environment [18]. Some of them contain natural products from secondary metabolic pathways that allow plants to protect themselves from their natural enemies [13]. In addition to many herbal plants occurring in the natural environment, certain pseudo-cereals are also rich in a wide range of compounds, e.g., flavonoids and phenolic acids, present not only in grains, but also in other aerial plant parts (leaves, stems) [19,20]. One such plant is *Amaranthus* spp. It is characterized by a high resistance to pathogens, and infection by pathogenic fungi does not pose a significant threat to plantations of this plant [21,22]. Plants exposed to stress, both biotic and abiotic, produce secondary metabolites such as phytohormones, betaine compounds, phenolic compounds, polysaccharides and oligosaccharides, fatty acids, sterols, humic acids and carotenoids. Literature reports indicate that these compounds can also act as plant growth stimulants [9,23]. 

The aim of the conducted research was a laboratory evaluation of the effect of amaranth leaf extracts on selected filamentous fungi occurring plants and the soil environment. The study used four species of amaranth from Poland and Turkey (*A. cruentus*, *A*. *hypochondriacus* × *hybridus*, *A. retroflexus*, *A. hybridus*). The biochemical composition of the extracts obtained from the leaves was analyzed, and their fungistatic activity against selected fungal species important in phytopathology and the food industry was assessed.

## 2. Results

### 2.1. Polyphenol Content

The research object was the leaves of *Amaranthus cruentus* (PC), *A. hypochondricus* × *hybridus* (PH), *A. retroflexus* (TR) and *A. hybridus* (TH). Dried amaranth leaves were used as a material for the production of plant extracts, in which the total polyphenol content and antioxidant activity were determined using the synthetic DPPH radical. Table 1 presents the results of the total polyphenol content in the tested extracts obtained from amaranth leaves and the respective antioxidant activity values. 

The highest concentration of total polyphenols in the initial extracts from amaranth leaves was recorded for *A. hybridus* (TH)—6.75 mg GAE/mL, while the lowest concentration of polyphenols was determined in the extract from *A. cruentus* (PC)—4.31 mg GAE/mL, which had an approx. 36% lower concentration of the tested active substances. It should be noted that extracts from plant material obtained from Turkish crops were characterized by significantly higher concentrations of polyphenols compared with amaranth extracts from Poland. The content of biologically active ingredients in plants is influenced by varietal factors, as well as cultivation factors (growing season, fertilization, soil, temperature, rainfall, etc.) [24]. Species of the genus *Amaranthus* are thermophilic plants. Turkey has more optimal growing conditions for this type of plant compared with Poland [25]. Optimal cultivation conditions promote more efficient synthesis of biologically active substances, which was confirmed by higher concentrations of polyphenolic compounds in the extracts of amaranth species cultivated in Turkey. The results of the antioxidant activity of the tested extracts corresponded with polyphenol levels. The highest antioxidant activity was observed in the extract obtained from the leaves of the species *A. hybridus* (TH) from Turkey (21.04 mM TE/mL), while extracts from Polish raw materials showed significantly lower antioxidant activity (13.84 mM TE/mL (PH) and 8.77 mM TE/mL (PC). 

### 2.2. Fungistatic Activity

Amaranth extracts exerted different effects on the growth of the tested fungal strains, depending on the fungal species, the type and concentration of the extract, and the duration of action of the biologically active ingredients (Table 2 and Table 3, Figure 1).

The fungistatic effects of amaranth leaf extracts largely depended on the fungus species tested (Table 2 and Table 3). The best results were recorded for *F. equiseti*, regardless of the concentration and type of extract (Table 3). Extracts from species cultivated in Poland such as *A. cruentus* (PC) and *A. hypochondriacus* × *hybridus* (PH) showed the strongest fungistatic effect, significantly inhibiting the surface growth of *F. equiseti* throughout the experiment (35.6–72.2%) (Figure 1). A strong fungistatic effect was also exhibited by the 15% extract from the species cultivated in Turkey, such as *A. retroflexus* (TR15), and the inhibition of fungal growth compared with the control remained at a high level throughout the experiment, i.e., 53.9–61.9% (Figure 1 and Figure 2). The extracts tested inhibited the growth of other species of the genus *Fusarium* to a low extent. The superficial growth of *F. solani* was most strongly inhibited by the 10% extract of *A. cruentus* (PC10), but only on day 4 of the experiment, and reached 49.4%. The remaining types and concentrations of extracts had no significant effect on *F. solani* growth inhibition; their antifungal activity in the first days of the experiment ranged from 1.3 to 26.5% and quickly decreased in the following days to a statistically insignificant level (2.4–11.8%), even contributing to a slight stimulation of the surface mycelium growth (TR10, TR15) (Figure 1). Amaranth extracts did not show any fungistatic activity against *F. oxysporum*, and the highest recorded level of fungal growth inhibition was 2.9–5.0% (TR10). Extracts, mainly from *A. hybridus* (TH10), stimulated superficial colony growth by up to 30% compared with the control sample. 

A strong fungistatic effect of amaranth extracts was recorded against *Rhizoctonia solani*, but only in the first days of the experiment (day 4) for all tested experimental combinations. Extracts from *A. cruentus* (PC15), *A. hybridus* and *A. retroflexus* (TH15, TR15) strongly reduced the surface growth of *R. solani* at the level of 66.3–80.8% (Figure 1 and Figure 2). The remaining extracts inhibited the growth of *R. solani* by only 37.8–44.7%. On subsequent days, the size of *R. solani* colonies exceeded the plate diameter (90.0 mm); therefore, measurements for individual experimental combinations were not performed (Figure 1). Amaranth extracts inhibited the growth of *A. alternata* only in the first days of the experiment, while they stimulated the growth of the fungus in the subsequent days. The strongest antifungal effect was recorded on day 4 of the experiment for extracts from species cultivated in Turkey: *A. retroflexus* (TH10—40.8%; TH15—55.4%) and *A. cruentus* (PC15—40.8%). Other concentrations of these extracts (PH, TR) inhibited *A. alternata* growth, but at a statistically insignificant level, i.e., 12.5–27.9%. In the consecutive days, the antifungal activity of the extracts decreased, and the surface growth of *A. alternata* was stimulated by more than 20% compared with the control sample (TR10) (Figure 1). Amaranth extracts, from species cultivated in both Poland and Turkey, showed very weak antifungal activity against *C. coccodes*. The highest degree of fungistatic effect was recorded only for the 15% *A. cruentus* extract at the beginning of the experiment (PC15—21.9%) and for the *A. hybridus* extract (TH10, TH15—20.4 and 26.9%, respectively), while their fungistatic effect in the following days rapidly decreased, even contributing to the stimulation of fungus growth. On the other hand, Amaranth extracts inhibited the surface growth of *T. harzianum*. Significant antifungal activity in the first days of the experiment was recorded for the following extracts: *A. hypochondriacus × hybridus* (PH10, PH15—37.4%; 37.8%) and *A. hybridus* (TH15—43.3%). In the following days of the experiment, the *T. harzianum* colony diameter exceeded the diameter of the plate (90.00 mm); therefore, measurements for individual experimental combinations were not performed (Figure 1)

The conducted experiment also focused on changes in the morphology of the fungi under the influence of plant extracts (Figure 2, Table 4). Most common were changes in the mycelial structure and the coloration of the obverse and reverse of the colony. The addition of extracts to the medium caused changes in the mycelial structure. Fungi growing on the medium with the addition of extracts formed less fluffy aerial mycelium with more relaxed growth. In some species, the disappearance of aerial mycelium and the growth of substrate mycelium were even observed (*C. coccodes*, *T. harzianum*). The reverse of the test fungi had a more intense color than in the control samples (*F. equiseti*, *F. oxysporum*) (Table 4). The addition of the extract to the medium caused deformation of conidia (*A. alternata*) and impaired sporulation. 

## 3. Discussion

Amaranth is a plant originating from Central America and is now widely cultivated in the countries of the tropics. In many countries, including India and tropical countries, it is still used in folk medicine as a laxative, to heal purulent lesions, boils and burns, and as an anti-malarial agent [26]. It has been proven that *Amaranthus*, thanks to its rich chemical composition, also shows antibacterial and antioxidant properties [27]. The plant is described in the literature as an important source of bioactive compounds such as lectins, phenols and flavonoids [28,29,30]. Flavonoids are an important group of biologically active compounds commonly present in many plant species. These compounds, contained in green plan parts, are characterized by, e.g., antimicrobial, anti-inflammatory or antioxidant activity [31,32]. The presence of twenty-five flavonoid and phenolic acids, such as protocatechuic acid, vanillic acid, gallic acid, salicylic acid, gentisic acid, *p*-hydroxybenzoic acid, β-resorcylic acid, syringic acid, ellagic acid, *m*-coumaric acid, *trans*-cinnamic acid, caffeic acid, chlorogenic acid, ferulic acid, sinapic acid, *p*-coumaric acid, rutin, naringenin, kaempferol, myricetin, catechin, isoquercetin, apigenin, hyperoside and quercetin, was detected in the leaves of *A. gangeticus*. Among the compounds, seven were identified as cinnamic acids, nine as benzoic acids and nine as flavonoid compounds. With respect to the three major classes of phenolics, the most prominent compounds were identified in four advanced lines of *A. gangeticus* genotypes in the following order: benzoic acids, cinnamic acids, flavonoids [33]. Not only flavonoids, but also phenolic acids determine the antimicrobial effect of amaranth extracts. The chemical composition of extracts varies and depends, among others, on the geographical location, as well as the composition and quality of the soil during plant growth [24]. The high content of polyphenolic compounds in alcoholic plant extracts was shown to be correlated with their high antioxidant capacity [34]. This was confirmed in our study involving amaranth leaf extracts carried out using the DPPH free radical method.

The biologically active compounds contained in amaranth plants are characterized by significant antimicrobial activity and cytotoxicity [35]. The conducted research showed the varied fungistatic effects of amaranth extracts. The strongest antifungal activity was recorded for extracts from *A. cruentus* (PC), *A. hypohondriacus* (PH) and *A. hybridus* (TH), of which TH was characterized by the highest polyphenol contents. The extracts strongly inhibited the growth of *F. equiseti*, *R. solani* and *T. harzianum*, and their highest fungistatic effect was observed at the beginning of the experiment. Jadhav and Biradar [20] conducted a similar study investigating the effect of *A. spinosus* on *Fusarium* spp. and *Aspergillus* spp. They showed that ethanol extract of amaranth leaves, applied at a concentration of 1000 µg/mL, strongly inhibited the surface growth of *F. oxysporum*, while its lower concentrations in the medium (100 µg/mL and 500 µg/mL) had a fungistatic effect only against *A. flavus* and *A. niger*. According to the studies conducted by Akbar et al. [36], *Amaranthus viridis* leaf extract demonstrated potent antifungal activity against *F. oxysporum,* because they reduced the fungal growth up to 48% in *F. oxysporum*. Our research did not confirm these reports, as it indicated not only a lack of inhibition of the surface growth of *F. oxysporum* (regardless of extract concentration), but even its stimulation. Studies on the antifungal effect of various plant extracts against *F. oxysporum* have been conducted by many researchers [12,17] who showed that *F. oxysporum* was not very sensitive to the effects of extracts from medicinal plants. Kursa et al. [13] showed that 20% ethanol extracts of tansy, yarrow and horseradish, considered potent against many phytopathogens, inhibited the surface growth of *F. oxysporum* only at the level of 24.7–26.56%, and only in the first days of the experiment. The strong inhibitory effect of the amaranth extract was recorded against *F. equiseti*, mainly in relation to the extracts of *A. retroflexus* (TR) and *A. hypochondriacus × hybrisus* (PH). *F. equiseti* is an important pathogen of crops, recorded increasingly often in horticultural plants that have been shown to be highly pathogenic to seedlings, causing root rot and decay in horticultural plants [8,37,38,39]. *F. equiseti*, as other species of the genus, produces toxins and antibiotics, especially trichothecenes and equisetin [1,40]. Therefore, in this regard, *A. cruentus* (PC), *A. hypochondriacus × hybrisus* (PH) and *A. retroflexus* (TR) can be considered as plants with significant antifungal potential. A study of Carminate et al. [41] confirmed the antifungal efficacy of *Amaranthus viridis* against *Colletotrichum musae* causing banana anthracnose. The present study showed no antifungal effect against *C. coccodes*, and a stimulating effect on mycelial surface growth was even observed. The growth of this species was also poorly inhibited by other plant extracts, including extracts from plants with high biocidal potential [13]. Carminate et al. [41] also indicated the great potential of *Amaranthus viridis* in controlling *F. solani* f. sp. *piperis*, responsible for fusariosis in black pepper. Similarly, our research demonstrated the possibility of using amaranth extracts (PC, PH, TH) to limit the development of *F. solani*. The strongest antifungal activity against the species was recorded for the *A. cruentus* (PC) extract at the beginning of the experiment. This raises the prospect of obtaining more interesting results using higher amaranth extract concentrations in the preventive plant protection against *F. solani*. Promising results were also obtained against *Rhizoctonia solani*. All the tested extracts strongly limited the surface growth of the fungus, but only in the first days of the experiment. Shirazi et al. [42] showed that *Amaranthus viridis* extract at concentrations of 1, 3 and 5% exhibited maximum inhibitory potential activity in the disc diffusion method against soil-borne pathogens *R. solani, F. oxysporum* and *M. phaseolina*. The authors reported that *A. viridis* could be used for the control of fungal diseases, particularly those caused by *R. solani*; promising results have also been obtained with *A. alternata*. Fungi of the genus *Alternaria* are pathogens of stored fruits and vegetables. They penetrate the plant through mechanical and enzymatic degradation of the cell wall, causing rotting in the stored crops. The destructive effect on the host plant is caused by the production of enzymes and specific (HST) and non-specific toxins (NHST) [6]. Higher concentrations of amaranth extracts (mainly TH and PC) already significantly inhibited the surface growth of *A. alternata* in the first days of the experiment, but their antifungal activity decreased over time. The obtained results were consistent with the study of Akbar et al. [36]. These authors showed that all organic solvent extracts from *Amaranthus viridis* significantly reduced the biomass of the tested fungi with increasing extract concentrations (5, 10, 15, 20 and 25 mg mL^−1^); strong activity was shown especially by ethyl acetate leaf fraction, resulting in reduced *A. alternata* growth by up to 44%.

Due to their cytotoxic properties, the biologically active compounds contained in plants have a direct effect on pathogen cells [35]. The antimicrobial activity of plant extracts rich in polyphenols causes many changes at the cellular level, e.g., damage of the microbial cell membrane through increased cell membrane permeability and, consequently, the leakage of cell contents [10]. Plant extracts from *Amaranthus* spp., in addition to inhibiting the growth of the tested fungi, caused changes in the color and structure of aerial mycelium, as well as sporulation. Kursa et al. [17] reported that the addition of plant extracts from tansy, sage and wormwood also caused changes in the color and structure of the aerial mycelium of the tested fungi. The results of the present experiment on amaranth extracts are a valuable source of information for further field research in plant protection against pathogens, and as an ingredient of biological products protecting agricultural crops from rotting.

## 4. Materials and Methods

### 4.1. Plant Materials

The leaves of fully matured *Amaranthus* spp. (BBCH19), *A. cruentus* (PC) and *A. hypochondriacus × hybridus* (PH) species from a cultivation located in Bodaczów near Zamość (south-eastern Poland) and *A. retroflexus* (TR) and *A. hybridus* (TH) from plantations located in the province of Düzce (north-western Turkey) were collected in 2019. The collected materials were dried at room temperature (in the shade) and subsequently stored in polyethylene bags at 4 °C.

### 4.2. Biochemical Characteristics of Extracts

#### 4.2.1. Extract preparation

Dried amaranth leaves were ground to a homogeneous fraction using an A11 Basic laboratory mill (IKA).

#### 4.2.2. Extraction

Crushed plant material (100.0 g) was weighed into round bottom flasks and then 1000 mL of 70% ethanol was added. Extraction was carried out for 6 h under reflux condenser at the boiling point of ethanol. The obtained extract was filtered through a sterile 22 µm filter (AlfaChem) and concentrated to 100 mL (1:1 extract) using a rotary evaporator (Heidolph Instruments, Schwabach, Germany). The final extract did not contain ethyl alcohol.

#### 4.2.3. Total Polyphenol Analysis

The concentration of total polyphenols in the tested extracts was determined using the spectrophotometric method (λ = 725 nm) with the Folin –Ciocalteau reagent, according to the modified method of Singelton and Rossi [43]. Phenol content results are expressed in gallic acid equivalents (GAE) (Sigma-Aldrich, St. Louis, MO, USA, ACS reagent ≥ 98.00%). The results were calculated based on the equation of the calibration curve prepared for gallic acid standards in the concentration range of 0.01–1 mg/mL (0.01, 0.02, 0.03, 0.04, 0.05, 0.06, 0.07, 0.08, 0.09 and 1.00 mg/mL). Each sample, depending on the initial concentration, was diluted according to the range on the standard curve [17]. All analyses were performed in triplicate.

#### 4.2.4. Assessment of Extract Antioxidant Activities

The antioxidant activity of plant extracts was determined using the modified method of Brand-Willams et al. [44] using the synthetic radical DPPH (1,1-diphenyl-2-picrylhydrase1 Sigma) converted to mM Trolox [45]. The inhibition of the DPPH radical by the extract sample was calculated according to the following formula: inhibition % = 100 (A_0_ − A_1_)/A_0_, where A_0_ is the absorbance of the control, and A_1_ is the absorbance of the sample. Each extract sample was diluted appropriately to the range of the standard curve prepared for Trolox (6-hydroxy-2,5,7,8-tetramethylchroman-2-carboxylic acid) standards. All analyses were performed in triplicate. Concentration values were based on the standard Trolox curve (0.2–1.2 mM) and expressed as millimoles of Trolox equivalents (TE) per ml of extract.

### 4.3. Fungal Cultures

Amaranth leaf extracts were individually tested against pathogenic fungi, such as *Fusarium solani* (ARIR14), *Fusarium oxysporum* (ECER4), *Fusarium equiseti* (ERIS8), *Alternaria alternata* (PCL10), *Colletotrichum coccodes* (P74/2), *Rhizoctonia solani* (TB71) and *Trichoderma harzianum* (A8/10). Fungal cultures were obtained in 2017–2020 as a result of plant mycological analyses (roots and aerial parts) of tomato (*Lycopersicon esculantum* L.) and pepper (*Capsicum annuum* L.) grown in the field. The fungal inoculum was derived from 10-day-old single-spore colonies grown on glucose-potato agar (PDA Difco, Becton, Dickinson and C., France), stored in the fungal collection of the Department of Plant Protection, University of Life Sciences in Lublin. Confirmation of strain species was carried out on the basis of microscopic analysis of each isolate/strain (spore structure and size, colony color) using appropriate mycological keys.

The study evaluated the effects of ethanol extracts of amaranth leaves (Polish and Turkish species: PC, PH, TR, TH) at concentrations of 10% and 15% on the linear growth of the test fungi. The method of poisoned substrates was used in the study, which is recommended to test chemical agents under laboratory conditions [13]. The method consisted of adding the test substance to sterile potato dextrose agar (PDA) medium cooled to 50 °C and inoculating the fungus species on the solidified medium. Medium and *Amaranthus* spp. extracts were poured into sterile Petri dishes of 90 mm and, subsequently, medium surface was inoculated with fungi colonies with a diameter of 3 mm. The control (CE) consisted of fungal colonies growing on potato dextrose agar (Difco PDA) with 10 and 15% residue after evaporation of the extractant used in the experiment (70% ethanol; a total volume of 1000 mL was evaporated to 100 mL in a rotary evaporator under the same conditions as in plant extract preparation). Five replicates of the tested extracts prepared at specific concentrations and added to each fungus were considered as objects. The plates prepared in this way were kept in an incubator for 12 days at 25°C. After 4, 8 and 12 days, the diameter of fungal colonies was measured. The measure of antifungal activity was the inhibition of mycelial growth on medium enriched with *Amaranthus* spp. extract relative to growth on control medium. The antifungal efficacy of *Amaranthus* spp. extract was calculated from the Abbott formula:I = [(C − T)/C] × 100% 
where: I—linear growth inhibition index of the fungus (percentage), C—diameter of the fungus colony in the control combination, T—diameter of the fungus colony in the combination containing the test substance concentration in the agar [13].

### 4.4. Statistical Analysis

Values are given as means ± standard deviation (SD) of each measurement. Where appropriate, the data were analyzed by analysis of variance (Duncan’s test) at the 5% significance level using the SAS statistical software (SAS Version 9.1, SAS Inst., Cary, NC, USA).

## 5. Conclusions

The use of natural compounds to control pathogens is very attractive and creates new opportunities for biological plant protection. The results of the present experiment demonstrated a very diverse effect of amaranth leaf extracts (*A. cruentus*, *A. hypochondriacus × hybridus*, *A. retroflexus*, *A. hybridus*) on selected phytopathogenic and antagonistic fungi. The extracts were fungistatic only against *F. equiseti*, *R. solani* and *A. alternata*, and showed activity only during the first days of the experiment, which proved their selective and short-term antifungal effect. The strongest fungicidal effect was recorded for species cultivated in Turkey such as *A. hybridus* (TH), which, due to having the highest content of polyphenols and high antioxidant activity, can be recommended for limiting the growth of some phytopathogenic fungi, also causing spoilage of agricultural crops. Due to the lack of a strong fungistatic effect, it can be used as an ingredient in plant preparations. In addition to direct antifungal activity, plant extracts act as elicitors of defense reactions in the plant and as biostimulants. The effect of amaranth leaf extracts as plant biostimulants requires further in-depth laboratory and field studies.

## Figures and Tables

**Figure 1 plants-12-01723-f001:**
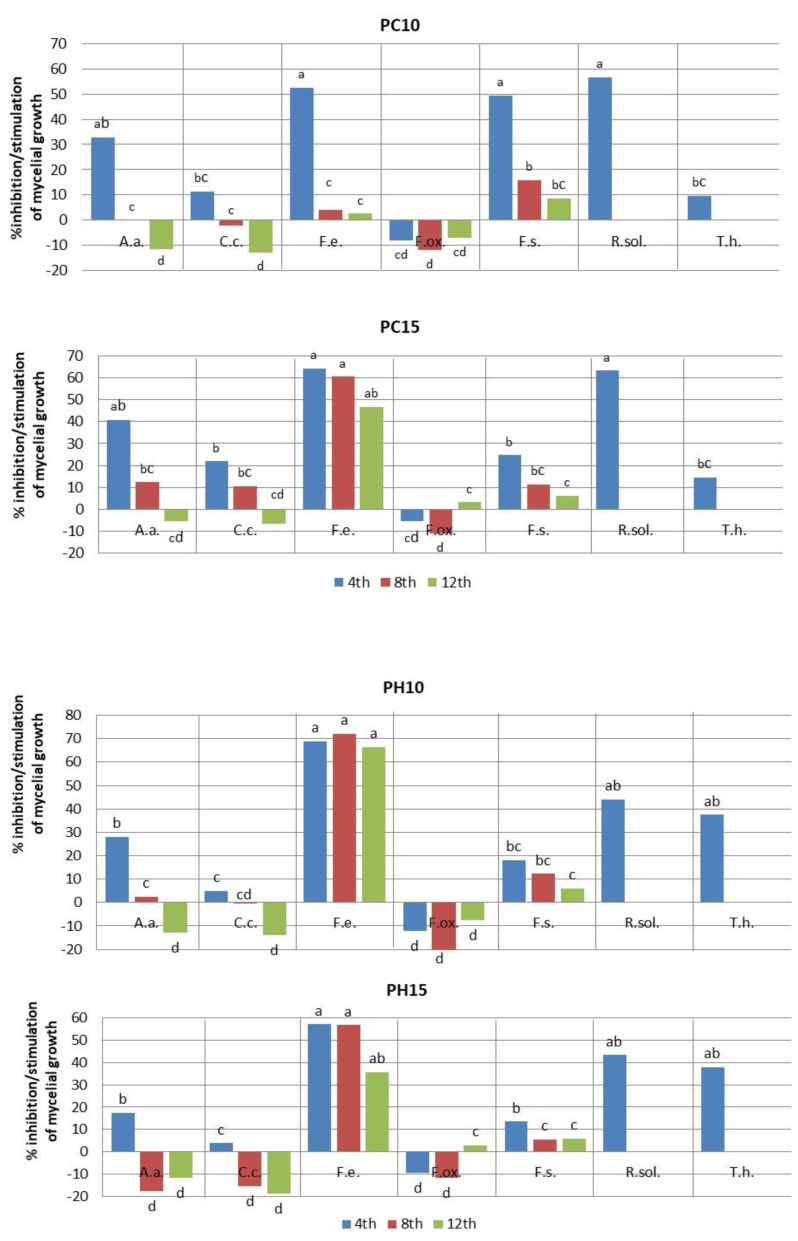
Inhibition/stimulation (%) of the mycelial growth after application of amaranth leaf extracts; PC—*A. cruentus* extract; PH—*A. hypochondriacus × hybridus* extract; TR—*A. retroflexus* extract, TH—*A. hybridus* extract; 10—extract concentration—10%; 15—extract concentration—15%; A.a.—*A. alternata*, C.c.—*C. coccodes*, F.e.—*F. equiseti*, F.ox.—*F. oxysporum*, F.s.—*F. solani*, R.sol.—*Rhizoctonia solani*, T.h.—*T. harzianum;* a–d—values marked with the same letter do not differ significantly at a significance level of *p* ≤ 0.05.

**Figure 2 plants-12-01723-f002:**
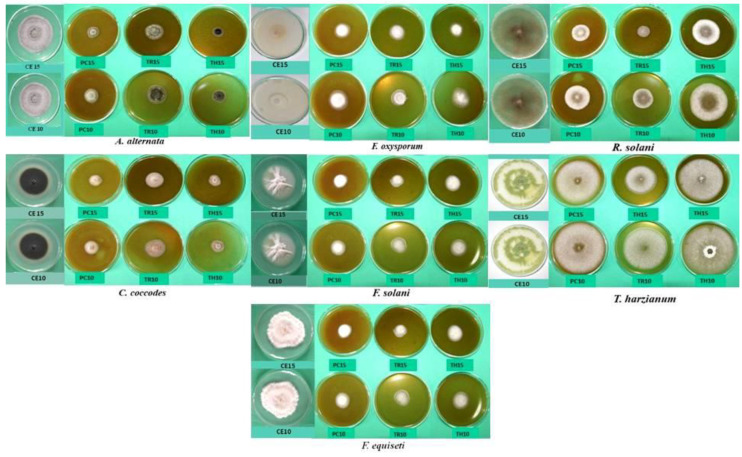
Linear growth of fungal colonies on PDA medium with the addition of amaranth leaf extracts (day 4). PC—*A. cruentus* extract; PH—*A. hypochondricus × hybridus* extract; TR—*A. retroflexus* extract, TH—*A. hybridus* extract; CE- control; 10—extract concentration—10%; 15—extract concentration—15%.

**Table 1 plants-12-01723-t001:** Total polyphenol content and antioxidant activity of amaranth extracts.

Plant Extract	Polyphenols (mg GAE/mL) ± SD	Antioxidant Activity, Free Radical Scavenging Ability	
		% Inhibition ± SD	mM TE/mL ± SD
*A. cruentus* (PC)	4.31 d ± 0.308	30.42 d ± 2.553	8.77 d ± 0.805
*A. hypochondriacus* × *A. hybridus L.* (PH)	5.39 c ± 0.259	46.51 c ± 0.225	13.84 c ± 0.071
*A. retroflexus* (TR)	5.81 b ± 0.082	56.59 b ± 1.137	17.02 b ± 0.359
*A. hybridus* (TH)	6.75 a ± 0.162	69.34 a ± 1.240	21.04 a ± 0.391

GAE—Gallic acid equivalent; TE—Trolox equivalent; SD—Standard deviation; a–d—Values in rows marked with the same letter do not differ significantly at the significance level of *p* ≤ 0.05.

**Table 2 plants-12-01723-t002:** Fungal colony diameter (mm) after application of amaranth extracts of *Alternaria alternata*, *Colletotrichum coccodes* and *Trichoderma harzianum*.

Experimental Combination	Concentration (%)	Number of Days ± SD		
		4	8	12
*Alternaria alternata*				
PC	10	23.0 ± 0.81 ef	51.0 ± 0.0 b	75.7 ± 1.62 ab
	15	20.3 ± 0.47 f	44.7 ± 2.05 d	71.7 ± 1.24 bc
PH	10	24.7 ± 0.47 de	49.7 ± 0.47 bc	76.7 ± 0.47 ab
	15	28.3 ± 0.81 bc	60.0 ± 0.0a	76.0 ± 0.47 ab
TR	10	30.0 ± 0.47 b	62.7 ± 0.47a	79.3 ± 0.47 ab
	15	26.3 ± 1.24 cd	60.7 ± 0.47 a	80.7 ± 0.47 a
TH	10	20.3 ± 0.47 f	47.7 ± 1.24 cd	67.0 ± +1.63 c
	15	15.3 ± 0.47 g	39.0 ± 0.81 e	59.0 ± 2.16 d
CE	10	34.3 ± 1.24 a	51.0 ± 0.81 b	68.0 ± 2.16 c
	15	34.3 ± 1.24 a	51.0 ± 0.81 b	68.0 ± 2.16 c
F		108.8507	132.152	37.9273
p		5.27 × 10^−15^	7.94 × 10^−16^	1.23 × 10^−10^
LSD		2.9	3.3	5.3
*Colletotrichum coccodes*				
PC	10	23.0 ± 1.24 cd	51.0 ± 0.0 bc	76.0 ± 0.47 ab
	15	20.3 ± 0.00 de	44.7 ± 0.47 d	71.7 ± 0.81 ab
PH	10	24.7 ± 0.81 bc	49.7 ± 0.47cd	76.7 ± 0.47 ab
	15	25.0 ± 0.00 bc	57.7 ± 0.47 a	80.0 ± 0.00 a
TR	10	28.3 ± 1.24 a	59.7 ± 0.47 a	78.7 ± 1.88 a
	15	26.7 ± 1.24 ab	55.0 ± 2.44 ab	79.3 ± 1.69 a
TH	10	20.7 ± 0.47 de	45.7 ± 2.86 de	72.7 ± 1.88 abc
	15	19.0 ± 0.81 e	44.0 ± 2.44 e	64.0 ± 7.78 c
CE	10	26.0 ± 0.81 abc	50.0 ± 0.00 bcd	67.3 ± 2.05 bc
	15	26.0 ± 0.81 abc	50.0 ± 0.00 bcd	67.3 ± 2.05 bc
F	29.8333	29.8333	32.84678	10.3675
p	1.12 × 10^−9^	1.12 × 10^−9^	4.64 × 10^−10^	8.72 × 10^−6^
LSD	0.7	3.1	5.1	10.0
*Trichoderma harzianum*				
PC	10	81.3 ± 0.47 c	* No measurements	
	15	77.0 ± 0.81 d		
PH	10	56.3 ± 0.47 g		
	15	56.0 ± 0.81 g		
TR	10	86.0 ± 0.81 b		
	15	72.3 ± 2.05 e		
TH	10	69.0 ± 0.81 f		
	15	51.0 ± 0.81 h		
CE	10	90.0 ± 0.00 a		
	15	90.0 ± 0.00 a		
F		527.6574	No correlations	
p		9.09 × 10^−22^		
LSD		3.2		

PC—*A. cruentus* extract, PH—*A. hypochondriacus × hybridus* extract; TR—*A. retroflexus* extract, TH—*A. hybridus* extract; CE—control; a–h—values in the rows marked with the same letter do not differ significantly at a significance level of *p* ≤ 0.05; LSD—the least significant difference; *—no measurements, fungal colony diameter exceeded plate diameter.

**Table 3 plants-12-01723-t003:** Fungal colony diameter (mm) after application of amaranth extracts of *Fusarium equiseti*, *Fusarium oxysporum*, *Fusarium solani* and *Rhizoctonia solani*.

Experimental Combination	Concentration (%)	Number of Days ± SD		
		4	8	12
*Fusarium equiseti*				
PC	10	13.3 ± 1.24 c	68.0 ± 2.16 a	86.7 ± 0.47 a
	15	10.0 ± 0.00 cd	28.0 ± 1.63 cde	47.3 ± 3.68 d
PH	10	8.7 ± 0.94 d	19.7 ± 0.47 e	30.0 ± 0.00 e
	15	12.0 ± 0.81 cd	30.7 ± 047 c	57.3 ± 2.05 c
TR	10	20.3 ± 1.24 b	43.7 ± 2.05 b	67.7 ± 6.12 b
	15	10.7 ± 0.47 cd	22.0 ± 1.63 de	41.0 ± 0.81 d
TH	10	28.0 ± 2.94 a	45.0 ± 5.31 b	85.0 ± 4.08 a
	15	20.0 ± 0.81 b	44.7 ± 0.47 b	70.7 ± 0.94 b
CE	10	28.0 ± 0.81 a	71.0 ± 2.94 a	89.0 ± 1.41 a
	15	28.0 ± 0.81 a	71.0 ± 2.94 a	89.0 ± 1.41 a
F		102.3873	140.2651	120.3965
P		9.56 × 10^−15^	4.43 × 10^−16^	1.97 × 10^−15^
LSD		4.4	8.7	9.9
*Fusarium oxysporum*				
PC	10	26.7 ± 2.35 b	60.0 ± 4.08 bc	75.0 ± 0.00 bc
	15	26.0 ± 0.00 b	59.7 ± 0.47 bcd	67.7 ± 2.05 d
PH	10	27.7 ± 1.69 ab	67.0 ± 0.00 a	75.3 ± 0.47 bc
	15	27.0 ± 2.16 b	60.0 ± 0.00 bc	68.0 ± 0.81 d
TR	10	25.3 ± 0.47 b	51.0 ± 0.00 d	71.7 ± 0.47 cd
	15	27.3 ± 0.81 b	53.7 ± 1.24 cd	77.0 ± 0.81 b
TH	10	32.2 ± 0.41 a	61.7 ± 3.68 ab	82.7 ± 2.05 a
	15	25.0 ± 0.00 b	57.0 ± 1.63 bcd	78.7 ± 1.24 ab
CE	10	24.7 ± 1.24 b	53.7 ± 1.24 cd	70.0 ± 1.36 d
	15	24.7 ± 1.24 b	53.7 ± 1.24 cd	70.0 ± 1.36 d
F		5.942576	12.40261	28.65224
P		0.000453	2.14 × 10^−6^	1.61 × 10^−9^
LSD		4.8	6.9	4.6
*Fusarium solani*				
PC	10	15.0 ± 0.00 e	52.7 ± 0.47 cd	77.7 ± 1.24 d
	15	22.3 ± 1.24 cd	55.7 ± 1.24 bc	79.7 ± 0.47 cd
PH	10	24.3 ± 0.47 bc	55.0 ± 2.44 bc	80.0 ± 0.00 cd
	15	25.7 ± 0.47 b	59.3 ± 0.94 ab	80.0 ± 0.00 cd
TR	10	30.3 ± 0.47 a	64.7 ± 0.47 a	81.7 ± 0.47 bc
	15	29.3 ± 0.47 a	64.7 ± 2.05 a	83.0 ± 1.63 ab
TH	10	24.0 ± 0.81 bc	41.3 ± 2.44 e	79.7 ± 0.47 cd
	15	21.7 ± 0.47 d	49.0 ± 0.00 d	75.0 ± 0.00 e
CE	10	29.7 ± 0.47 a	62.7 ± 1.69 a	85.0 ± 0.00 a
	15	29.7 ± 0.47 a	62.7 ± 1.69 a	85.0 ± 0.00 a
F		122.9346	22.87798	40.0
P		1.61 × 10^−15^	1.21 × 10^−8^	7.52 × 10^−11^
LSD		2.2	5.6	2.5
*Rhizoctonia solani*				
PC	10	39.0 ± 0.81 c	*—No measurements	
	15	30.3 ± 1.24 c		
PH	10	50.3 ± 0.47 b		
	15	51.0 ± 0.47 b		
TR	10	56.0 ± 3.26 b		
	15	49.7 ± 6.12 b		
TH	10	30.3 ± 0.47 c		
	15	17.3 ± 2.05 d		
CE	10	90.0 ± 0.00 a		
	15	90.0 ± 0.00 a		
F		158.4078	No correlation	
P		1.34 × 10^−16^		
LSD		9.5		

PC—*A. cruentus* extract; PH—*A. hypochondriacus × hybridus* extract; TR—*A. retroflexus* extract, TH—*A. hybridus* extract; CE—control; a–e—values in the rows marked with the same letter do not differ significantly at a significance level of *p* ≤ 0.05; LSD—the least significant difference; *—no measurements, fungal colony diameter exceeded plate diameter.

**Table 4 plants-12-01723-t004:** Selected features of fungal morphology under the influence of plant extracts (day 8 of the experiment).

Fungus Species	Experimental Combination	Mycelium Surface and Structure	Obverse	Reverse	Presence of Spores
*A. alternata*	PC10, PC15	Aerial, regular	White and gray	Black	No aleuroconidia
	PH10, PH15	Aerial, regular	White and gray	Black	No aleuroconidia
	TH10, TH15	Aerial, regular, fluffy	Gray	Black	Sparse or no aleuroconidia
	TR10, TR15	Poor growth, low mycelium slightly compacted, regular	Grey and black; gray	Black	Numerous deformed aleuroconidia
	CE10, CE15	fluffy, regular growth	Gray	Black	Sparse aleuroconidia
*C. coccodes*	PC10, PC15	Substrate, regular	White and salmon; microsclerotia in the center	Colorless	Sparse conidia
	PH10, PH15	Substrate, regular	White and salmon; no microsclerotia	Colorless	No conidia
	TH10, TH15	Aerial, regular	White and orange; sparse black microsclerotia in the center	Colorless	No conidia
	TR10, TR15	Aerial, regular	White and orange; sparse black microsclerotia in the center	Colorless	No conidia
	CE10, CE15	Substrate	Light white; sparse black microsclerotia	Colorless	Sparse conidia
	PC10, PC15	Substrate, restricted aerial	White; green sporulation on the edge	Colorless	Very numerous conidia
	PH10, PH15	Substrate, restricted aerial	White; green sporulation in the center	Colorless	Very numerous conidia
*T. harzianum*	TH10, TH15	Substrate, restricted aerial	White; high green sporulation	Colorless	Very numerous conidia
	TR10, TR15	Substrate, restricted aerial	White; high dark green sporulation	Colorless	Very numerous conidia
	CE10, CE15	Aerial, regular	White; light green sporulation	Colorless	Sparse conidia
	PC10, PC15	Aerial, regular	White and creamy; creamy and beige	Colorless; creamy and brown	Moderately numerous conidia
	PH10, PH15	Aerial, regular	White and creamy; creamy and beige	Colorless; creamy and brown	Moderately numerous conidia
*F. equiseti*	TH10, TH15	Abundant aerial mycelium	Creamy and white	Colorless	Sparse conidia
	TR10, TR15	Mainly substrate, restricted aerial	Creamy and white	Colorless	Sparse conidia
	CE10, CE15	Aerial, regular	Creamy and white	Colorless	Numerous microconidia and sparse macroconidia
	PC10, PC15	Regular growth	White	Colorless	Medium-sized microconidia
	PH10, PH15	Regular growth	White; pink and white	Colorless	Medium-sized microconidia
*F. oxysporum*	TH10, TH15	Regular growth	White	Purple and red	Microconidia
	TR10, TR15	Regular growth	White	Purple and red	Microconidia
	CE10, CE15	Regular growth	White	Colorless	Macro- and microconidia
*F. solani*	PC10, PC15	Aerial, regular, abundant	White	Colorless	No conidia
	PH10, PH15	Aerial, regular, abundant	White	Colorless	No conidia
	TH10, TH15	Aerial fine, substrate	White	Creamy	Sparse conidia
	TR10, TR15	Aerial fine, substrate	White	Creamy	Sparse conidia
	CE10, CE15	Aerial regular	White	Colorless	Very numerous conidia
*R. solani*	PC10, PC15	Aerial, fluffy, abundant	White	Colorless	-
	PH10, PH15	Aerial, fluffy, abundant	White and creamy	Colorless	-
	TH10, TH15	Aerial, fluffy, abundant	White and creamy	Colorless	-
	TR10, TR15	Aerial, fluffy, abundant	White and creamy	Colorless	-
	CE10, CE15	Aerial, fluffy, abundant	Creamy and brown	Creamy	-

PC—*A. cruentus* extract; PH—*A. hypochondriacus × hybridus* extract; TR—*A. retroflexus* extract, TH—*A. hybridus* extract; 10—extract concentration 10%; 15—extract concentration 15%; day 8.

## Data Availability

All data included in the main text.

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
