# Peer review of "Antifungal Potency of Amaranth Leaf Extract: An In Vitro Study"

_plants, 2023, doi:10.3390/plants12081723_

Round 1
Reviewer 1 Report
The authors have addressed the major issue in the food industry that of plant diseases. They studied the use of phytochemicals from the plant source i.e., Amaranth spps on the filamentous fungi which target plants in the environment and are also present in the soil. They used extracts from four species of amaranth (A. cruentus, A. hypochondriacus 80 x hybridus, A. retroflexus, A. hybridus) against fungal species i.e., Alternaria alternata, Colletotrichum coccodes, Trichoderma harzianum, Fusarium species and Rhizoctonia solani. The overall study provides data on how this plant species can be used as an antifungal agent against a variety of fungal species. They have shown that different leaf extracts inhibit fungal growth to different extents. The experiments are well performed and conclusions are supported by well-documented data. However, the presentation of the data is still not convincing.
Comments:
1. Give a brief description of the experiment in the 2.1 results section before you mention the results. (which also includes the abbreviation you are using for different extracts).
2. In tables control is mentioned as CE but in legends it is EC. Change it
3. Mention what LSD is in the table legends.
4. Figure 1 Try to make the graphs in different colours for easy differentiation. For non-significant values, I would recommend using ns rather than other abbreviations. And for significant differences use Asterix.
5. Figure 2 is missing the growth pattern in PH extracts (mentioned in the legend). Fungal colonies of F. equiseti are also missing.
6. Did the authors check the growth in the liquid medium? If not adding that data would definitely make the study more impactful.
7. Have they tried to mix the leaf extracts of two species against any fungal species?
Author Response
Review 1
Thanks for detailed review 1 and valuable comments on the text.
Point 1. Give a brief description of the experiment in the 2.1 results section before you mention the results. (which also includes the abbreviation you are using for different extracts).
Brief description of the experiment in the results section 2.1. has been added to clarify the abbreviations used in the text.
Point 2. In tables control is mentioned as CE but in legends it is EC. Change it
In the legend of tables 1-2, the abbreviation for the control combination was corrected instead of EC to CE.
Point 3. Mention what LSD is in the table legends.
The legend below the tables explains the meaning of the abbreviation LSD
Point 4. Figure 1 Try to make the graphs in different colours for easy differentiation. For non-significant values, I would recommend using ns rather than other abbreviations. And for significant differences use Asterix.
Figure 1 has been corrected to make it more visible: color differences are marked. In the case of statistical differences, letter symbols have not been retained because it is difficult to mark statistical differences on the graph using the abbreviation “ns” or Asterisks, as suggested by the reviewer, so in this case no changes were made, because in our opinion such marking is also correct.
Point 5. Figure 2 is missing the growth pattern in PH extracts (mentioned in the legend). Fungal colonies of F. equiseti are also missing.
Figure 2 was corrected according to the reviewer's comments, adding photos of control combinations (CE) and photos of F. equiseti
Point 6. Did the authors check the growth in the liquid medium? If not adding that data would definitely make the study more impactful.
The tests were carried out on a solid medium - PDA potato glucose agar; explanation in the methodology
Point 7. Have they tried to mix the leaf extracts of two species against any fungal species?
The extracts were not mixed during the experiment because there are a preliminary study. Phytotron tests on model plants will be carried out using a mixture of the most effective amaranth extracts.

Reviewer 2 Report
The present manuscript “Antifungal potency of amaranth leaf extract: an in vitro study” demonstrated the antifungal activity of leaf extracts of four amaranth species (A. cruentus, A. hypochondriacus x hybridus, A. retroflexus and A. hybridus) against selected strains of fungi. The results suggested that the antimicrobial properties of the tested extracts varied depending on the amaranth species and the fungal strain. Identification of new and sustainable products for plant protection against diseases is need of the time. The manuscript demonstrated in vitro studies against the selected strains of fungi, which is very preliminary data in my opinion. I would suggest including a detached leaf essay where it can be clearer if extracts have some antifungal activities. Additionally the extracts can be evaluated against the spore germination rate in different time intervals.
With Regards
Author Response
Review 2
Thanks for detailed review 2 and valuable comments.
I would suggest including a detached leaf essay where it can be clearer if extracts have some antifungal activities. Additionally the extracts can be evaluated against the spore germination rate in different time intervals.
The conducted research is a preliminary test that shows whether amaranth leaf extracts have antifungal properties. The reviewer's suggestions are broader activities for a new research part using live plant leaves and amaranth extracts. Such research will be done at a later time as another experiment. Phytotron tests on model plants will be carried out using a mixture of the most effective amaranth extracts.

Round 2
Reviewer 2 Report
Dear Authors
Thank you for addressing the suggestions and hereby say that I do not have any further queries.
Regards